# CAREL: Instruction-guided reinforcement learning with cross-modal auxiliary objectives

**Armin Saghafian,** *Sharif University of Technology*     *armin.saghafian@gmail.com*

**Amirmohammad Izadi,** *Sharif University of Technology*     *amirmohammad.izadi01@sharif.edu*

**Negin Hashemi Dijujin ,** *Sharif University of Technology*     *n.hashemi94@sharif.edu*

**Mahdieh Soleymani Baghshah,** *Sharif University of Technology*     *soleymani@sharif.edu*

**Reviewed on OpenReview:** *https://openreview.net/forum?id=zJUEYr5X1X*

## Abstract

Grounding the instruction in the environment is a key step in solving language-guided goal-reaching reinforcement learning problems. In automated reinforcement learning, a key concern is to enhance the model's ability to generalize across various tasks and environments. In goal-reaching scenarios, the agent must comprehend the different parts of the instructions within the environmental context in order to complete the overall task successfully. In this work, we propose **CAREL** (*Cross-modal Auxiliary REinforcement Learning*) as a new framework to solve this problem using auxiliary loss functions inspired by video-text retrieval literature and a novel method called instruction tracking, which automatically keeps track of progress in an environment. The results of our experiments suggest superior sample efficiency and systematic generalization for this framework in multi-modal reinforcement learning problems. Our code base is available here.

## 1 Introduction

Numerous studies have examined the use of language goals or instructions within the context of reinforcement learning (RL) Röder et al. (2021); Geffner (2022); Luketina et al. (2019). Language goals typically provide a higher-level and more abstract representation than goals derived from the state space Rolf & Asada (2014). While state-based goals often specify the agent's final expected goal representation Liu et al. (2022); Eysenbach et al. (2022), language goals offer more information about the desired sequence of actions and the necessary subtasks Liu et al. (2022). Therefore, it is important to develop approaches that can extract concise information from states or observations and effectively align it with textual information, a process referred to as grounding Röder et al. (2021).

Previous research has attempted to ground instructions in observations or states using methods such as reward shaping Goyal et al. (2019); Mirchandani et al. (2021) or goal-conditioned policy/value functions Zhong et al. (2019); Hejna III et al. (2021); Akakzia et al. (2020); Deng et al. (2020), with the latter being a key focus of many studies. Their approaches incorporate various architectural or algorithmic inductive biases, such as cross-attention Hanjie et al. (2021), hierarchical policies Jiang et al. (2019); Andreas et al. (2017), and feature-wise modulation Madan et al. (2021); Chevalier-Boisvert et al. (2018). Typically, these works involve feeding instructions and observations into policy or value networks, extracting internal representations of tokens and observations at each time step, and propagating them through the network. Previous studies have explored auxiliary loss functions to improve these internal representations in RL Stooke et al. (2021); Wang et al. (2023); Zheng et al. (2023), and have emphasized the importance of self-supervised/unsupervised learning objectives Levine (2022) in RL. However, these loss functions lack the alignment property between different input modalities, such as visual/symbolic states and textual commands/descriptions. Recent studies

have suggested contrastive loss functions to align text and vision modalities in an unsupervised manner Ma et al. (2022); Yao et al. (2021); Radford et al. (2021); Yu et al. (2022); Li et al. (2022). Most of these studies fall under the video-text retrieval literature Zhu et al. (2023); Ma et al. (2022), where the language tokens and video frames align at different granularities. Since these methods require a corresponding textual input along with the video, the idea has not yet been employed in language-informed reinforcement learning, where the sequence of observation might not always match the textual modality (due to action failures or inefficacy of trials). One can leverage the success signal or reward to detect the successful episodes and consider them aligned to the textual modality containing instructions or environment descriptions. Doing so, the application of the above mentioned auxiliary loss functions makes sense.

Contrastive loss serves as a fundamental mechanism in representation learning, particularly in scenarios where distinguishing between similar and dissimilar objects is essential. As outlined in Radford et al. (2021) , this loss function operates by minimizing the distance between corresponding items in two modalities, such as sequences of observations and their corresponding instructions, while simultaneously maximizing the distance between unrelated or dissimilar items. This loss enhances representations via utilizing the weak supervision that is available as pairs of matched observation sequence and instruction from the successful trajectories and unmatched pairs from unsuccessful ones.

In this study, we propose a new framework, called **CAREL** (*Cross-modal Auxiliary REinforcement Learning*), for the adoption of auxiliary grounding objectives from the video-text retrieval literature Zhu et al. (2023), particularly X-CLIP Ma et al. (2022), to enhance the learned representations within these networks and improve cross-modal grounding at different granularities. By leveraging this grounding objective, we aim to improve the grounding between language instructions and observed states by transferring the multi-grained alignment property of video-text retrieval methods to instruction-following agents. We also propose a novel method to mask the accomplished parts of the instruction via the auxiliary score signals calculated for the cross-modal loss while the episode progresses. This helps the agent to focus on the remaining parts of the task without repeating previously done sub-tasks or being distracted by past goal-relevant entities in the instruction. Our experiments on the MiniGrid and BabyAI environments Chevalier-Boisvert et al. (2018) showcase the idea's effectiveness in improving the systematic generalization and sample efficiency of instruction-following agents. The primary contributions of our work are outlined as follows:

- We designed an auxiliary loss function to improve cross-modal alignment between language instructions and environmental observations.

- We introduced a novel instruction tracking mechanism to help the agent focus on the remaining tasks by preventing the repetition of completed sub-tasks.

- We enhanced overall performance and sample efficiency in two benchmarks.

## 2 Methods

Language-Informed Reinforcement Learning (LIRL) builds upon traditional RL by integrating natural language instruction as a structured input to enrich the learning process. In our proposed method, we aim to leverage the instruction more effectively, going beyond merely conditioning the model on it. We introduce a novel loss function that enhances representation learning by incorporating a contrastive loss between the observation and instruction modalities, thereby achieving more meaningful and grounded representations for observations. The enhanced representation not only streamlines the reinforcement learning process but also enables tracking of the instruction and dynamic conditioning of the model on the remaining components of the instruction throughout the trajectories.

In this study, we first incorporate an auxiliary loss inspired by the X-CLIP model Ma et al. (2022) to enhance the grounding between instruction and observations in instruction-following RL agents. This auxiliary loss serves as a supplementary objective, augmenting the primary RL task with a multi-grained alignment property, which introduces an additional learning signal to guide the model's learning process. This design choice was motivated by the ability of contrastive loss to effectively create meaningful embeddings through aligning

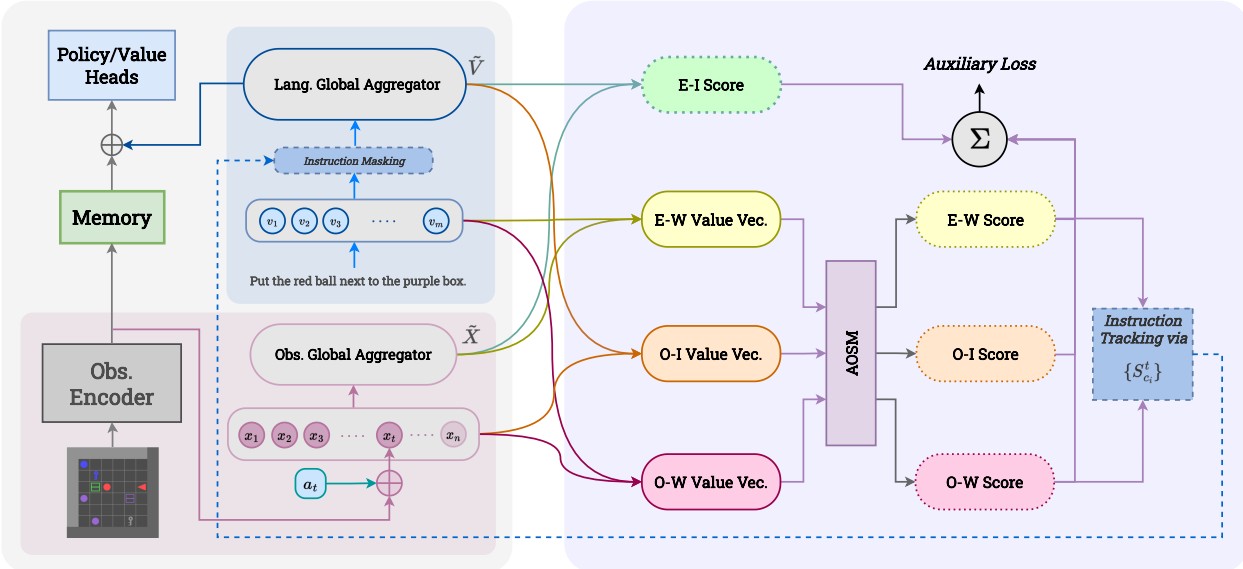

Figure 1: **Overall view of CAREL.** In this figure, we showcase CAREL over a candidate baseline model from Chevalier-Boisvert et al. (2018). (*Left*) The blue box handles the (masked) instruction and its local/global representations, while the pink box contains the components related to observation. (*Right*) The purple box shows the calculation steps for the X-CLIP loss and tracks scores for instruction masking.

observations with the intended instruction, ultimately enhancing the overall performance of the RL system. Additionally, we leverage the alignment scores calculated within the X-CLIP loss to track the accomplished sub-tasks and mask their information from the instruction. This masking aims to filter out the distractor parts of the instruction and focus on the remaining parts, improving the overall sample efficiency of the agents. We call this technique *instruction tracking*. Both the proposed auxiliary loss and the instruction tracking ability can act as frameworks and don't alter the structure of the base model. In the remainder of this section, we explain the auxiliary loss and the instruction tracking separately.

## 2.1 Auxiliary Loss

The auxiliary loss in CAREL is applied only to successful trajectories to ensure that the loss signals align with goal-relevant behavior. During training, a batch of trajectories is collected, each containing observations, actions, rewards, and corresponding instructions. The aggregated rewards for each trajectory are calculated to determine success. A fraction of the maximum achievable reward is used as the acceptance threshold, ensuring that only sufficiently high-performing trajectories are classified as successful.

This separation is done only for the auxiliary loss, and the base model's RL loop is run over all interactions, whether successful or unsuccessful. Hence, it differs from offline RL in which only certain episodes are selected for the whole training process Levine et al. (2020).

Each successful episode contains a sequence of observation-action pairs $ep = ([O_1, a_1]...[O_n, a_n])$ meeting the instructed criteria and an accompanying instruction $instr = (I_1, ..., I_m)$ with $m$ tokens. Since the X-CLIP loss requires local and global encoders for each modality, we must choose such representations from the model or incorporate additional modules to extract them. To explore the exclusive impact of the auxiliary loss and minimize any changes to the architecture, we use the model's existing observation and instruction encoders, which are crucial components of the model itself. We utilize these encoders to extract local representations for each observation-action $[O_t, a_t]$ denoted as $x_t \in \mathbb{R}^{d \times 1}$, $t = 1, ..., n$ in which each action is embedded similar to positional embedding in Transformers Vaswani et al. (2017) and is added to the observation representation. Each instruction token $I_i$ is encoded as $v_i \in \mathbb{R}^{d \times 1}$, $i = 1, ..., m$. The global representations can be chosen from the model itself or added to the model by aggregation techniques such as mean-pooling or attention. We denote the global representations for observations and the instruction by $\tilde{x}$ and $\tilde{v}$, respectively. The

auxiliary loss function is then calculated according to Ma et al. (2022) as below. We restate the formulas in our context to make this paper self-contained.

To utilize contrastive loss, we first need to calculate the similarity score for each episode ($ep$), a sequence of observations, and an instruction ($instr$) pair denoted as $s(ep, instr)$. To do this, we calculate four separate values; Episode-Instruction ($S_{E-I}$), as well as Episode-Word ($S_{E-W}$), Observation-Instruction ($S_{O-I}$) and Observation-Word ($S_{O-W}$) similarity values. Episode-Instruction score can be calculated using this formula:

$$S_{E-I} = \tilde{x}^T \tilde{v}, \tag{1}$$

with $\tilde{x}, \tilde{v} \in \mathbb{R}^{d \times 1}$, $S_{E-I} \in \mathbb{R}$. Other values are calculated similarly:

$$S_{E-W} = (V\tilde{x})^T, \tag{2}$$

$$S_{O-I} = X\tilde{v}, \tag{3}$$

$$S_{O-W} = XV^T, \tag{4}$$

where $X = (x_1^T; ...; x_n^T) \in \mathbb{R}^{n \times d}$ is local representation for observations, $V = (v_1^T; ...; v_m^T) \in \mathbb{R}^{m \times d}$ in local representations for instruction tokens, and $S_{E-W} \in \mathbb{R}^{1 \times m}$, $S_{O-I} \in \mathbb{R}^{n \times 1}$ and $S_{O-W} \in \mathbb{R}^{n \times m}$ provides fine-granular similarities between the language instruction and the episode of observation. These values are then aggregated with appropriate attention weights via a technique called **A**ttention **O**ver **S**imilarity **M**atrix (**AOSM**). Episode-Word ($S'_{E-W}$) and Observation-Instruction ($S'_{O-I}$) scores are calculated from the values as follows:

$$S'_{O-I} = Softmax(S_{O-I}[., 1])^T S_{O-I}[., 1], \tag{5}$$

$$S'_{E-W} = Softmax(S_{E-W}[1, .])^T S_{E-W}[1, .], \tag{6}$$

where:

$$Softmax(x[.]) = \frac{\exp(x[.]/\tau)}{\sum_j \exp(x[j]/\tau)}, \tag{7}$$

in which, $\tau$ controls the softmax temperature. For the Observation-Word score, bi-level attention is performed, resulting in two fine-grained similarity vectors. These vectors are then converted to scores similar to the previous part:

$$S'_{instr}[i, 1] = Softmax(S_{O-W}[i, .])^T S_{O-W}[i, .], \qquad i \in \{1, ..., n\}, \tag{8}$$

$$S'_{ep}[1, i] = Softmax(S_{O-W}[., i])^T S_{O-W}[., i] \qquad i \in \{1, ..., m\}, \tag{9}$$

where $S'_{instr} \in \mathbb{R}^{n \times 1}$ show the similarity value between the instruction and $n$ observations in the episode and $S'_{ep} \in \mathbb{R}^{1 \times m}$ represents the similarity value between the episode and $m$ words in the instruction.

The second attention operation is performed on these vectors to calculate the Observation-Word similarity score ($S'_{O-W}$):

$$S'_{O-W} = (Softmax(S'_{ep}[1, .])^T S'_{ep}[1, .] + Softmax(S'_{instr}[., 1])^T S'_{instr}[., 1])/2. \tag{10}$$

The final similarity score between an episode and an instruction is computed using the previously calculated scores:

$$s(ep, instr) = (S_{E-I} + S'_{E-W} + S'_{O-I} + S'_{O-W})/4. \tag{11}$$

This method takes into consideration both fine-grained and coarse-grained contrasts. Considering $N$ episode-instruction pairs in a batch of successful trials, the auxiliary loss is calculated as below:

$$\mathcal{L}_{aux} = -\frac{1}{N} \sum_{i=1}^{N} (\log \frac{\exp(s(ep_i, instr_i))}{\sum_{j=1}^{N} \exp(s(ep_i, instr_j))} + \log \frac{\exp(s(ep_i, instr_i))}{\sum_{j=1}^{N} \exp(s(ep_j, instr_i))}) \tag{12}$$

The total objective is calculated by adding this loss to the primary RL loss, $\mathcal{L}_{RL}$, with a coefficient of $\lambda_C$.

$$\mathcal{L}_{total} = \mathcal{L}_{RL} + \lambda_C.\mathcal{L}_{aux} \tag{13}$$

The overall architecture of a base model Chevalier-Boisvert et al. (2018) and the calculation of the auxiliary loss are depicted in Figure 1. If the shape of the output representations from the observation and instruction encoders does not align, we employ linear transformation layers to bring them into the same feature space. This transformation is crucial as it facilitates the calculation of similarity between these representations within our loss function.

---

**Algorithm 1** CAREL framework

---

1: Initialize baseline $\pi_\theta(a \mid s, I)$ and env
2: Get instruction $I$ from the environment
3: **for** each batch of rollouts **do**
4:     **for** each rollout in batch **do**
5:         **while** episode not done **do**
6:             $a_t \sim \pi_\theta(a_t \mid s_t, I)$
7:             Execute $a_t$, and Recieve $r_t$, $s_{t+1}$
8:             Baseline stores $(s_t, I, a_t, r_t, s_{t+1})$ for it's own loss
9:             $s_t \leftarrow s_{t+1}$
10:             **if** episode terminated and successful **then**
11:                 Save the whole episodes: $\mathcal{B}_{\text{success}} \leftarrow ep_i$
12:             **end if**
13:         **end while**
14:     **end for**
15:     Update $\theta$ based on baseline's RL loss
16:     Update $\theta$ using $\mathcal{B}_{\text{success}}$ to compute and apply the auxiliary loss ($\mathcal{L}_{aux}$) based on Eq. 12
17: **end for**

---

## 2.2 Instruction Tracking

We can consider the similarities from eqs. 1 to 4 as a measure of matching between the instruction and the episode at different granularities. Once calculated at each time step of the episode, this matching can signal the agent about the status of the sub-task accomplishments. The agent can then be guided toward the residual goal by masking those sub-tasks from the instruction. More precisely, at time step $t$ of the current episode, the agent has seen a partial episode $ep^{(t)} = ([O_1, a_1], ..., [O_t, a_t])$ that in a fairly trained model should align with initial stages of the instruction. The instruction itself can be parsed into a set of related sub-tasks $C = \{c_i\}$ via rule-based heuristics, and there can be constraints on their interrelations. However, if prior knowledge of the instruction is unavailable, we can leverage Natural Language Processing tools such as SpaCy or Language Models to carry out this process. For example, an instruction of the form *"Do X, then do Y, then do Z"* includes three sub-tasks $X$, $Y$, and $Z$ which have a sequential order constraint $(X \rightarrow Y \rightarrow Z)$. Other examples could involve different forms of directed graphs where a specific sub-task is *acceptable* only if its parents have been satisfied before during the episode. The set of acceptable sub-tasks at time step $t$ is denoted by $C_t$, which contains the root nodes in the dependency graphs at the start of the episode.

In order to track the accomplished sub-task, we assess the similarity between $C$ members and the partial episode. This can be done by tracking $S_{E-W}$ or $S_{O-W}$, which provides fine-grained similarities across the language modality. In the case of $S_{E-W}$, the similarity per token in $c_i$ is averaged to get a final scalar

similarity. For $S_{O-W}$, the maximum similarity between the observations and each word is considered for averaging across $c_i$ tokens. Another option is to calculate a learned representation for the whole $c_i$ instead of averaging and to track the instructions based on their similarity with the partial episode, aiming at preserving the contextual information in the representation of the sub-task. The final calculated similarity of each acceptable sub-task $c_i$, denoted by $S_{c_i}^t$, is tracked at each time step. Once this similarity rises significantly, the matching is detected, and $c_i$ is removed from the instruction participating in the language-conditioned model. More precisely, we remove $c_i$ from the instruction when the following condition is satisfied:

$$(c_i \in C_t) \quad \wedge \quad (S_{c_i}^t \geq k \times \frac{1}{t-1} \sum_{j=1}^{t-1} S_{c_i}^j). \tag{14}$$

Here, $k > 1$ is a hyperparameter that specifies the significance of the matching score's spike. While the auxiliary loss described in the previous subsection is applied on the episode level, instruction tracking happens at every time step of the episode over the partial episode and the masked instruction. This process is represented in Figure 1.

These two techniques can be applied jointly, as the auxiliary loss improves the similarity scores through time, and the improved similarities enhance the instruction tracking. To prevent false positives during tracking at the initial epochs of training, one can constrain the probability of masking and relax this constraint gradually as the learning progresses.

---

**Algorithm 2** Instruction Tracking (IT) framework

---

1: **Input:** Environment $env$ and instruction $I$
2: Split instruction $I$ into sub-tasks by rule-based heuristics ($C = \{c_i\}$)
3: **for** $t$ in $max\_steps$ **do**
4:     $o_t = env.step()$
5:     Compute each sub-task's ($c_i$) similarity with this step's observation ($o_t$)
6:     $S_{c_i}^t \leftarrow mean(V_{c_i} \tilde{x}_t)$
7:     Keep moving average of $S_{c_i}$ over time in $\tilde{S}_{c_i}$
8:     **if** $S_{c_i}^t > \tilde{S}_{c_i} \times k$ **then**
9:         with a probability $p$:
10:             $I \leftarrow$ Omit detected sub-task ($c_i$) from the instruction
11:             $C \leftarrow$ Omit detected sub-task ($c_i$) from the $C$
12:     **end if**
13:     Give the new $I$ to the policy
14: **end for**

---

### 2.2.1 Instruction Tracking Implementation Details

**Splitting into subtasks**
To implement Instruction Tracking, the subtasks need to be extracted from the initial instruction. In our case, the environments provide a clear and consistent instruction format, which enables us to split sentences into individual words by using string matching to identify conjunctions for subtask separation. We tokenize the instructions and determine the positions of conjunctions. This process produces a list of subtasks, each paired with its corresponding conjunction.

For example, the "GoToSeq" environment uses the following structure:

"go to a/the {color} {type}" + "and go to a/the {color} {type}" + ", then go to a/the {color} {type}" + "and go to a/the {color} {type}"

Given this format, we tokenize the sentence and match the words "and" and "then" to identify and separate the subtasks.

Example:

```
"Go to the red box and go to a green ball, then go to the blue key."
```

We use the conjunctions "and" and "then" to split the instruction into the following subtasks:

```
"Go to the red box" + and + "go to a green ball" + then + "go to the blue key."
```

**Masking Process**
To exclude a completed subtask, we mask both the subtask tokens and their associated conjunction by replacing these tokens with a `<mask>` token when reconstructing the instruction. This idea, inspired by sequential thinking, helps the model avoid repeating tasks by letting it focus on the remaining tasks when a portion of the tokens is masked.

If the condition in 14 is met for any of the subtasks—for example, the first one—we mask that subtask with a probability. The final instruction will then look like this:

```
"<mask> <mask> <mask> <mask> <mask> <mask> go to a green ball, then go to the blue key."
```

This masked instruction is then passed to the model.

To extend this for other environments without a clear format, NLP tools such as SpaCy or Large Language Models like GPT-4o can be used to break the instructions down to subtasks and subsequently to mask the completed ones.

**Masking Probability.**
When the threshold in Equation 14 is reached, we only apply the masking with a certain probability. This number starts very low in the initial stages of training and rises over time following a tanh function. We implement this to avoid falsely masking uncompleted subtasks at the early stages of training, when the encoders haven't been properly trained and don't produce strong representations.

The exact function for the probability is as follows:

$$\text{prob} = \tanh\left(\frac{\text{current\_frame}}{\text{max\_frames}}\right)$$

where max\_frames determines the total number of training steps.

## 3 Experiments

In our experiments, we conducted a comparative analysis to assess the impact of X-CLIP Ma et al. (2022) auxiliary loss on generalization and sample efficiency of instruction-following agents. We showcase the success of CAREL along with the instruction tracking technique in our experiments[1]. For this purpose, we employ a baseline called BabyAI Chevalier-Boisvert et al. (2018) (the proposed model along with the BabyAI benchmark), for which we explain the experimental setup and results in the following paragraphs. Additionally, to compare our results with recent works, we consider SHELM Paischer et al. (2023) , which uses CLIP embedding to detect objects in the observation, and LISA Garg et al. (2022) , which leverages imitation learning for a sample efficient training process.

### 3.1 CAREL Results

We employ the BabyAI environment Chevalier-Boisvert et al. (2018), a lightweight but logically complex benchmark with procedurally generated difficulty levels, which enables in-depth exploration of grounded language learning in the goal-conditioned RL context. We use BabyAI's baseline model as the base model and minimally modify its current structure. Word-level representations are calculated using a simple token

---

[1]For the experiments reported in this paper, we have used one NVIDIA 3090 GPU and one TITAN RTX GPU over two weeks.

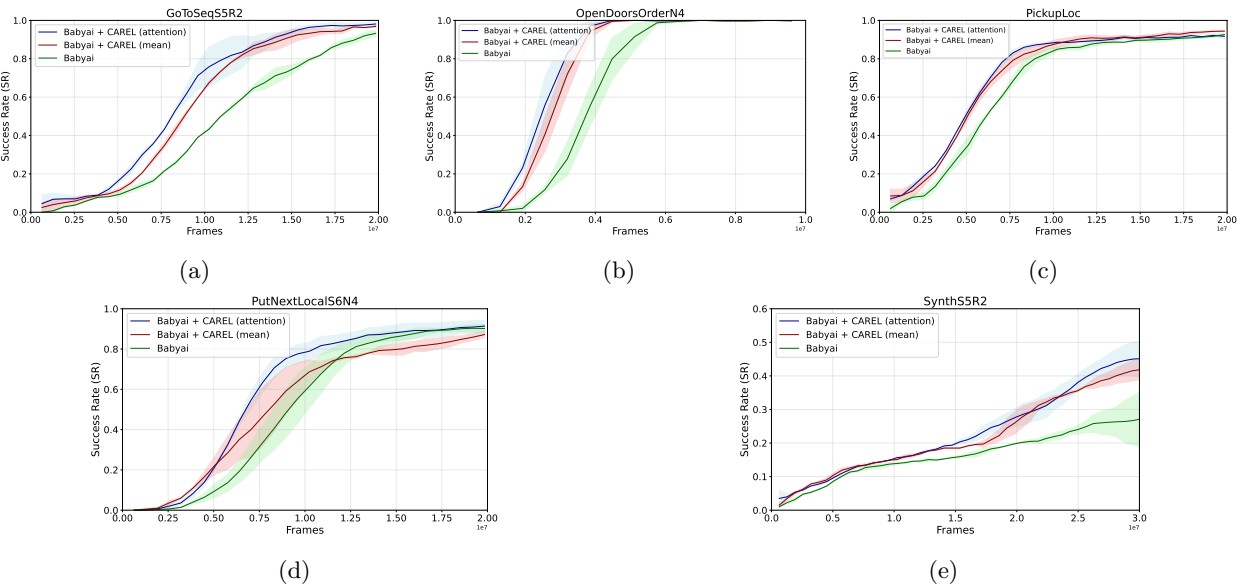

Figure 2: Test SRs indicating the overall effect of Vanilla CAREL on BabyAI. (The results are smoothed before plotting.)

embedding layer. Then, a GRU encoder calculates the global instruction representation. Similarly, we use the model's default observation encoder, a convolutional neural network with three two-dimensional convolution layers. All observations pass through this encoder to calculate local representations. Mean-pooling/Attention over these local representations is applied as the aggregation method to calculate the global observation representation.

The evaluation framework for this work is based on systematic generalization to assess the language grounding property of the model. We report the agent's success rate (SR) over a set of unseen tasks at each BabyAI level, separated by pairs of color and type of target objects or specific orders of objects in the instruction. This metric is recorded during validation checkpoints throughout training.

Figure 2 illustrates the improved sample efficiency brought about by CAREL auxiliary loss (without instruction tracking and action embedding to minimize the modifications to the baseline model, hence called *Vanilla CAREL*). All results are reported over two random seeds. The results indicate improved sample efficiency of CAREL methods across all levels, especially those with step-by-step solutions that require the alignment between the instruction parts and episode interactions more explicitly, namely `GoToSeq` and `OpenDoorsOrder`, which contain a sequence of `Open`/`GoTo` subtasks described in the instruction. The generalization is significantly improved in more complex tasks, i.e., `Synth`.

### 3.1.1 Sample Efficiency

Figures 2 and 3 present a comparison between the sample efficiency of our method and other baselines. In addition to this, we have conducted a quantitative study. Table 1 highlight the superiority of our method's sample efficiency, as we achieve better performance with the same number of samples. The reported values roughly represent the halfway point of the samples required for model convergence, except for the Synth environment, which did not converge due to its complexity.

### 3.1.2 CAREL + Instruction Tracking

For instruction tracking, we use only the $S_{E-W}$ vector and average over tokens of each sub-task to track the score over time. To detect sub-task matching from the score signal, we set $k = 2$ in Equation 14. All the other settings are kept the same as in vanilla CAREL, except that we also add action embeddings to local observation representations, as described in the Auxiliary Loss section. We mask acceptable sub-tasks with

| Method | Task | Samples | Success Rate |
|---|---|---|---|
| Babyai | GoToSeqS5R2 | 10M | 41% |
| Babyai + CAREL | | 10M | 73% |
| Babyai | OpenDoorsOrderN4 | 3M | 24% |
| Babyai + CAREL | | 3M | 79% |
| Babyai | PickupLoc | 5M | 35% |
| Babyai + CAREL | | 5M | 53% |
| Babyai | PutNextLocalS6N4 | 7.5M | 34% |
| Babyai + CAREL | | 7.5M | 63% |
| Babyai | SynthS5R2 | 20M | 20% |
| Babyai + CAREL | | 20M | 28% |

Table 1: Impact of CAREL on sample efficiency.

a certain possibility that follows a hyperbolic tangent function in terms of training steps. This is meant to minimize the amount of masking at the start of the learning process when the model has not yet learned a good embedding for instructions and observations, and increase it over time.

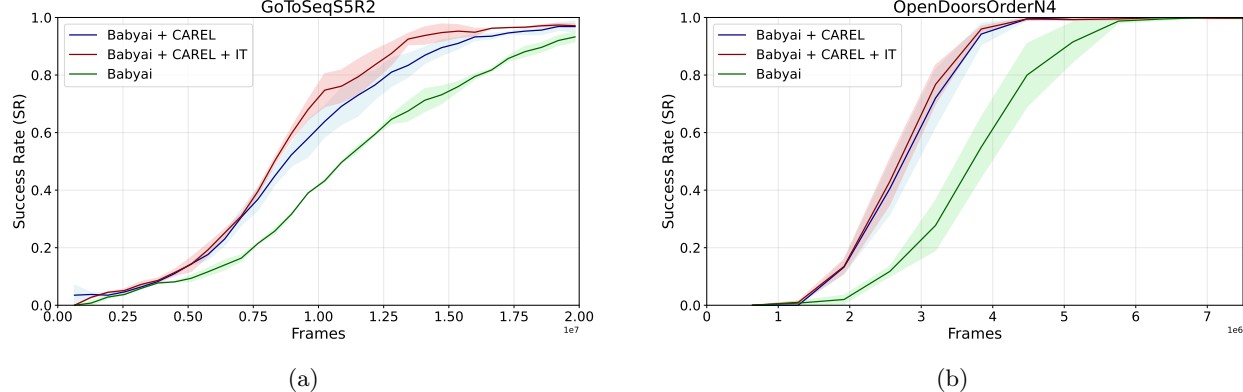

Figure 3: Test SRs indicating the effect of CAREL + instruction tracking on the baseline models. (The results are smoothed before plotting.)

The results of the full CAREL method (with instruction tracking and action embedding) are reported on the `GoToSeq` and `OpenDoorsOrder` environments. We break down the instructions in this environment with a rule-based parsing to increase the level of detail in the instructions. The instruction, stated initially as `"open the [color1] door, then open the [color2] door"`, is converted to `"go to the [color1] door, then open the [color1] door, then go to [color2] door, then open the [color2] door"` and so on. This introduces the challenge of sequential sub-tasks into BabyAI tasks. The results in Figure 3 indicate that instruction tracking improves CAREL. This improvement is more apparent in the case of environments with multistep, complex tasks that include more intermediate sub-tasks, such as `GoToSeq`.

## 3.2 Comparison with Recent Models

To compare our model against recent baselines, we apply and evaluate our models using two baselines, SHELM Paischer et al. (2023) and LISA, Garg et al. (2022). We evaluate the effectiveness of our framework by applying it on top of SHELM while comparing the performance of LISA with the BabyAI + CAREL model.

To evaluate the capability of our framework on RGB environments, we apply and test it on SHELM Paischer et al. (2023). SHELM leverages the knowledge hidden in pre-trained models such as CLIP and Transformer-XL. It also uses CLIP to extract textual tokens related to every observation. Then, these tokens are passed through the frozen Transformer-XL network to form a memory of tokens throughout the episode. This hidden memory is then concatenated to a CNN representation of observation and passed to actor/critic heads. We must modify SHELM's structure as it doesn't use the environment's instructions, which are crucial to success in a multi-goal setting. To do so, we utilize BERT's tokenizer to embed the instructions and pass them through a Multihead-Attention layer with four heads. The resulting embedding is concatenated to the hidden layer alongside the outputs of the CNN model and Transformer-XL, which are then passed to the actor-critic head. We consider the encoder output for observations as the local representations and add another Multi-head Attention layer followed by a mean-pooling over them to calculate the corresponding global representations. Table 2 , shows the sample efficiency of our approach by reporting the success rates for each model. The number of frames is chosen as the midpoint of convergence. Additionally, Figure 4 demonstrates faster convergence of our framework compared to the SHELM baseline.

We conducted further experiments to evaluate our model against imitation learning, which is known for its fast training and sample efficiency. Table 3 details the comparison in samples and success rate between our models. We present the results for the LISA and BabyAI + CAREL models on the number of frames required for their convergence. Additionally, we provide the results for BabyAI under the same number of frames as BabyAI + CAREL to demonstrate our sample efficiency. The results for LISA represent the highest performance that this model can achieve whereas our model achieves significantly higher performance upon convergence.

Please refer to the Appendix for more results on time consumption A.1, coefficient tuning, and ablation studies on Instruction Tracking A.2.

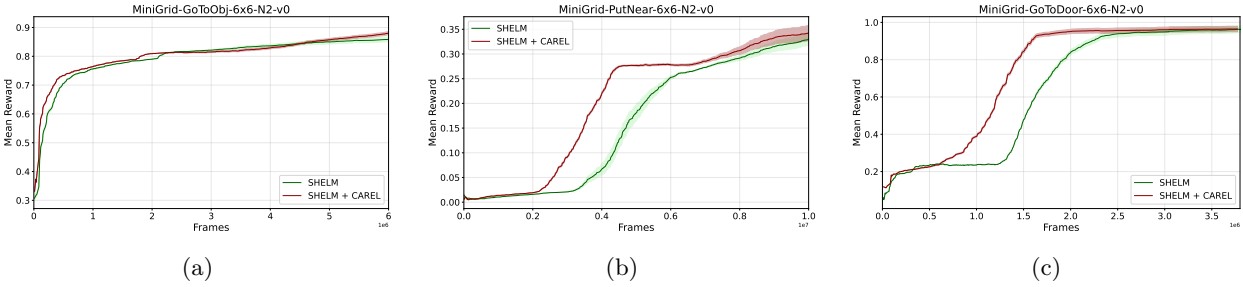

(a)  (b)  (c)

Figure 4: Sample efficiency comparison after applying the CAREL framework to the SHELM baseline. (The results are smoothed before plotting.)

| Model | Task | Frames | Success Rate |
|---|---|---|---|
| SHELM | PutNear | 5M | $20\% \pm 3\%$ |
| SHELM + CAREL | | 5M | $27\% \pm 2\%$ |
| SHELM | GoToDoor | 1.5M | $50\% \pm 3\%$ |
| SHELM + CAREL | | 1.5M | $83\% \pm 1\%$ |
| SHELM | GoToObject | 1.5M | $77\% \pm 1\%$ |
| SHELM + CAREL | | 1.5M | $78\% \pm 2\%$ |

Table 2: Sample efficiency and performance comparison of CAREL on SHELM baseline.

| Model | Task | Frames | Success Rate |
|-------|------|--------|--------------|
| BabyAI | | 14M | $76\% \pm 3\%$ |
| LISA | GoToSeqS5R2 | 7M (100k trajectories) | $77\% \pm 2\%$ |
| BabyAI + CAREL | | 14M | $93\% \pm 2\%$ |
| BabyAI | | 4M | $62\% \pm 5\%$ |
| LISA | OpenDoorsOrderN4 | 1.5M (100k trajectories) | $60\% \pm 3\%$ |
| BabyAI + CAREL | | 4M | $97\% \pm 2\%$ |

Table 3: Sample efficiency and performance comparison with LISA. (LISA converges on 100k trajectories, and an increase in the number of frames will not improve performance.)

## 4  Limitations

CAREL framework is designed to improve the model's understanding of common concepts in observation and instruction via representation learning. As a result, it can only be applied to environments that provide clear textual instruction to the agent. The framework does not impose any constraint on the environment or the MDP beyond providing instructions.

The primary requirement for applying CAREL is that the base model must generate embeddings for visual observations and textual instructions via separate encoders (e.g. dual-encoder models) or another mechanism.

## 5  Conclusion

This paper proposes the CAREL framework which adopts auxiliary cross-modal contrastive loss functions to the multi-modal RL setting, especially instruction-following agents. The aim is to improve the multi-grained alignment between different modalities, leading to superior grounding in the context of learning agents. We apply this method to existing instruction-following agents. The results indicate the sample efficiency and generalization boost from the proposed framework. As for the future directions of this study, we suggest further experiments on more complex environments and other multi-modal sequential decision-making agents. Also, the instruction tracking idea seems to be a promising direction for further investigation.

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

## A  Appendix

**Implementation Details**

**Base Models.** The BabyAI environment serves as a standard benchmark for instruction-following reinforcement learning. Its base model is trained using the PPO algorithm  Schulman et al. (2017)  and Adam optimizer with parameters $\beta_1 = 0.9$ and $\beta_2 = 0.999$. The learning rate is $7e-4$, and the batch size is 256. We set $\lambda_C = 0.01$ and the temperature $\tau = 1$ as CAREL-specific hyperparameters. To minimize the changes to the baseline model updates, we backpropagate the gradients in an outer loop of PPO loss to be able to capture episode-level similarities. This gradient update with different frequencies has been tried in the previous literature Madan et al. (2021).

The actor-critic model from the SHELM model was also used as a baseline. We train the learnable parts of the model using the PPO algorithm and Adam optimizer with the same hyperparameters. The learning rate is $1e-4$, and the batch size is set to 16.

### A.1   Time Consumption

To evaluate the CAREL framework's time consumption, we designed an experiment to measure the frames per second (FPS) achieved by our method and the baseline across various BabyAI benchmark environments. We calculated the average FPS over 100,000 frames detailed in Table 4. The FPS results provide an indication of our framework's computational efficiency and time consumption compared to the baseline. Based on the results obtained across different tasks in the BabyAI environment, our method processes approximately 420 frames (observations) per second on average, compared to the 668 FPS achieved by the BabyAI baseline. This translates to an approximate runtime of 13 hours for processing 20 million frames (as commonly presented in the tables as the maximum step count), compared to 8 hours for the BabyAI baseline. However, this does not provide a substantial challenge, as the main issue in Reinforcement Learning is the number of interactions with the environment needed for training.

Many works on RL Yu (2018); Ladosz et al. (2022) consider the number of interactions between the agent and the environment—referred to as sample efficiency—as the primary challenge, rather than the overall training time. This is because interactions with the environment are inherently constrained by the environment's properties and cannot be accelerated by increasing computational resources.

| Environment | Average-FPS (Babyai) | Average-FPS (CAREL) |
|---|---|---|
| GoToSeqS5R2 | 629.60 | 512.19 |
| OpenDoorsOrderN4 | 690.80 | 641.50 |
| PickupLoc | 679.86 | 265.33 |
| PutNextLocalS6N4 | 688.75 | 287.51 |
| SynthS5R2 | 655.66 | 394.32 |
| **Average** | 668.93 | 420.17 |

Table 4: Comparison of Average Frames Per Second (FPS) Between Babyai and CAREL Across Multiple Environments.

### A.2   Ablations

### A.2.1   Auxiliary Loss Coefficient

We evaluated the impact of auxiliary loss coefficient ($\lambda$) on performance of CAREL framework. As shown in Fig 5 values above 0.01 or below 0.001 result in a sudden decline in performance with 0.01 achieving the highest performance.

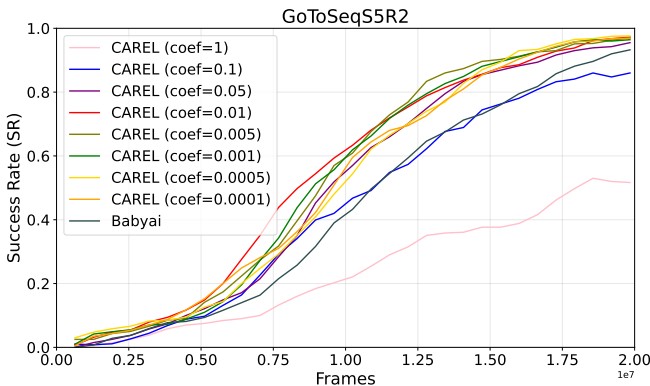

Figure 5: Experiment on the impact of the coefficient for our auxiliary loss.

### A.2.2 Action Embeddings in Instruction Tracking

Instruction Tracking (IT) can use actions taken up to this point to predict the completion of a subtask; therefore, adding the action embeddings to the observations helps the model to better detect the completion of subtasks that are utilized in the IT mechanism. To further investigate this, we conducted a CAREL+IT experiment on the `GoToSeqS5R2` environment, comparing performance with and without action embeddings. The results are depicted in Figure 6.

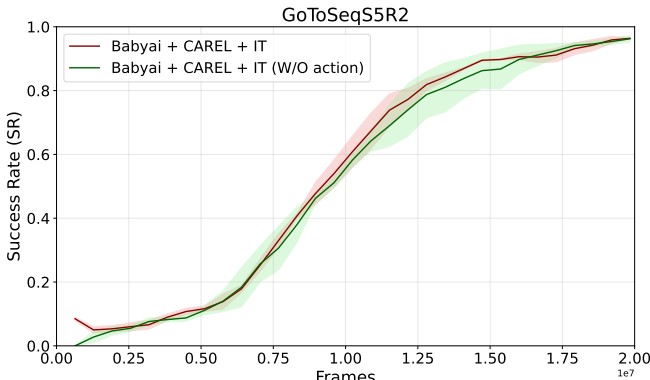

Figure 6: Effect of adding action embeddings to the observations in Instruction Tracking.

### A.2.3 Instruction Tracking Similarity Score

When calculating the threshold 14, we can only use the $S_{E-W}$ or $S_{O-W}$ because we need the similarity of each word separately to be able to calculate a final score for each subtask. Between the two, we only used $S_{E-W}$ in our experiments for computational efficiency. The experiment in Figure 7 indicates that the $S_{E-W}$ works as well as $S_{O-W}$. We employ $S_{E-W}$ because it is more computationally efficient due to its lower dimension.

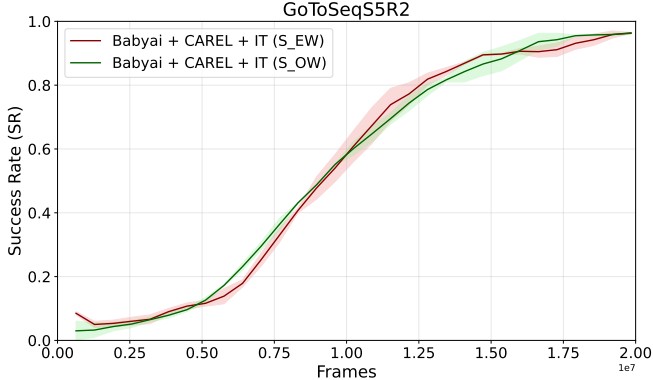

Figure 7: Results for using different embeddings for calculating the similarity score in Instruction Tracking

### A.2.4 Impact of CAREL in Instruction Tracking

This experiment shows the effect of CAREL on the improvement achieved by applying Instruction Tracking. As shown in Figure 8, applying Instruction Tracking to our model (baseline + CAREL) results in a slightly higher improvement compared to its application on the original baseline.

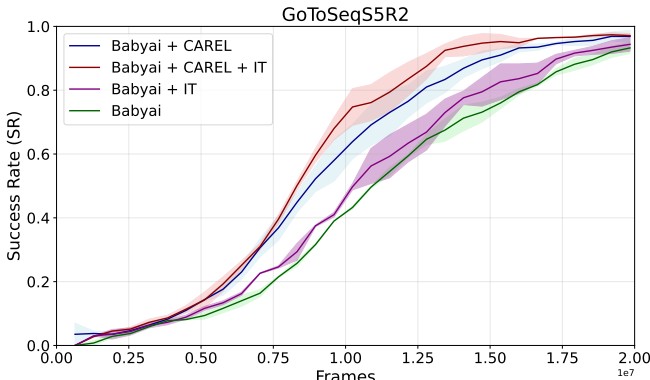

Figure 8: Ablation study on the impact of combining CAREL with Instruction Tracking.

### A.3 Environments

**MiniGrid Environment.** MiniGrid is designed for research in reinforcement learning and decision-making tasks. It provides a grid-based world where agents interact with their surroundings by perceiving symbolic or RGB observations and executing discrete actions. The environment is fully customizable, allowing researchers to define unique grid layouts, object types, and interaction rules. MiniGrid supports tasks such as navigation, object manipulation, and goal-directed behaviors, offering varying levels of complexity to suit different experimental needs.

**BabyAI Environment.** BabyAI is an environment based on the MiniGrid environment. It is specifically crafted to assess the performance of agents in following natural language instructions. BabyAI provides a range of procedurally generated tasks that require agents to achieve goals by acting on textual commands. These tasks are diverse, challenging agents in areas like navigation and interaction with objects.

The complexity of tasks in BabyAI varies significantly. Basic tasks involve simple objectives such as moving to a specific location (e.g., "Go to the green square") or retrieving an object (e.g., "Pick up the red key"). More intricate tasks demand multiple interactions, such as "go to a red ball then go to a blue box".

### A.4 Preliminary

**Reinforcement Learning.** Our experiments were conducted in two environments modeled as partially observable Markov decision processes (POMDPs), represented by the tuple $(S, A, O, \Omega, P, \gamma, R)$. In this framework $s \in S$ represents the states of the environment, $a \in A$ denotes the actions, $o \in \Omega$ refers to observations drawn from the observation model $O(o|s, a)$, $P(s'|s, a)$ defines the transition dynamics, $R$ is the reward function, and $\gamma$ is the discount factor. The objective during training is to determine a policy $\pi$ that maximizes the expected total discounted reward, expressed as:

$$\max_{\pi} \mathbb{E}_{\pi} \left[ \sum_{t=0}^{\infty} \gamma^t R(s_t, a_t) \right].$$

