# OpenReview forum: "CAREL: Instruction-guided reinforcement learning with cross-modal auxiliary objectives"
_TMLR — Accepted by TMLR_

### Review · Reviewer_3YgW · 2025-06-01

**Summary Of Contributions:**

This paper proposed CAREL and instruction tracking to improve instruction-guided reinforcement learning models. CAREL serves as an auxiliary training objective to improve the internal observation and instruction representations. Instruction tracking keeps track of the progress of the task completion and remove the completed sub-tasks from the instruction. In the experiments on two benchmarks, the proposed methods are shown to be more performant and sample efficient than the baseline method.

**Audience:**

Yes

**Broader Impact Concerns:**

No ethical concerns.

**Claims And Evidence:**

Yes

**Requested Changes:**

1. It would be appreciated if the authors could clarify the details mentioned in Weakness 3.
2. It would be appreciated if the paper could compare the performance of the proposed method with some of the previous methods mentioned in Section 1.
3. In Equation (14), a hyperparameter $k$ is required. Although the authors set $k=2$ in the experiments, it would be appreciated if the authors could discuss the impact of these hyperparameter. Especially, would this hyperparameter be task-dependent?
4. When plotting Figure 2 to show the data efficiency of the proposed method, the x-axis is number of frames. However, as Table 2 suggests, the proposed method has a lower FPS than the baseline. It would be appreciated if the paper could show a performance comparison plot with the running time as x-axis to further demonstrate the efficiency.

Typos:
1. In Equation (12), $1/n$ should be $1/N$.

**Strengths And Weaknesses:**

## Strengths:
1. The proposed methods, CAREL and instruction tracking, are both well-motivated and can be easily plugged into existing instruction-guided RL frameworks.
2. In the experiments, the proposed methods improve the performance comparing to the baseline method. Especially, the proposed method significantly improves the sample efficiency of the RL models. With the same numbers of training steps, the methods with the proposed CAREL and instruction tracking always yield better or on-par performance than the baseline.
3. The paper provides pseudo code of the proposed methods, which help the readers to understand the pipeline.


## Weaknesses:
1. Although the proposed methods improve the sample efficiency, the final performance is not always improved. For example, in PickupLoc and PutNextLocalS6N4 in Figure 2, and MiniGrid-GoToDoor-6x5-N2-v0 in Figure 5, the final performance is just on-par or even below the baseline.
2. As the authors mentioned in the second paragraph of Section 1, there are a few prior works exploring instruction-guided RL and internal representation improvement in RL. However, the paper does not compare the performance of the proposed method with any of these works.
3. There are a few technical details remain unclear:
+ In the instruction tracking method, the paper mentions that instruction can be parsed into a set of sub-tasks via rule-based heuristics. However, it is unclear how the rule-based heuristics are implemented. Some examples would be helpful for the readers to understand.
+ In the instruction tracking method, the algorithm need to remove sub-tasks from the instruction. As the instruction is in natural language, how is this achieve? Is there any NLP tool used here?
+ It is unclear how frequently is the auxiliary loss calculated. Is it computed after every one rollout of an episode? Providing a pseudo code about the entire RL framework would be helpful to clarify this point.
+ In Section 3.1.4, the method uses "a certain possibility that follows a hyperbolic tangent function in terms of training steps". What is exactly the expression of this function?

---

> ### Author Response · Authors · 2025-06-30
>
> We sincerely thank you for your detailed and constructive review.
>
> Weakness 1:
>
> As shown in Fig. 2, the final performance of our method (the attention variation) is generally on par or better than the baseline. Overall, our model improves the final performance along with sample efficiency in more complicated environments that feature many objects of various shapes and colors, which is the primary focus of our image-text alignment loss function. For example, the SynthS5R2 environment that reaches a much lower success rate even after training on more frames shows a clear advantage over the baseline. In contrast, due to the simplicity of the previously mentioned environments, the baseline eventually achieves near-perfect performance given sufficient training, so our approach does not result in significant improvements in the final performance of those cases, even though it still greatly improves on sample efficiency.
>
> Weakness 2:
>
> Our current RL baselines, BabyAI [1] and Semantic Helm (SHELM) [2], both fall under the instruction-guided RL and internal representation improvement. We introduced BabyAI in the second paragraph of the Introduction section, but did not reiterate it later when discussing these two groups. Both these baselines train embedding modules to improve representations fed into the policy.
>
> Weakness 3:
> We have provided additional explanation and examples for your information in the new implementation details subsection under the instruction tracking Section 2.2 that covers all the questions. However, we still reiterate them here for your convenience:
>
> Weakness 3.1:
>
> In our case, the environments provide a clear and consistent instruction format, which enables us to split sentences into individual words by using string matching to identify conjunctions for subtask separation.
> For instance, the “GoToSeq” environment uses the following structure:
>
> “go to a/the {color} {type}” + “and go to a/the {color} {type}” + “, then go to a/the {color} {type}” + “and go to a/the {color} {type}”
>
> Given this format, we tokenize the sentence and match the words “and” and “then” to identify and separate the subtasks.
> To extend this for environments without a clear format, Large Language Models such as GPT-4o can be used to break the instructions down to subtasks.
>
> Weakness 3.2:
>
> We tokenize the instructions and identify the positions of conjunctions to divide them into subtasks since the instructions have a clear format. This process produces a list of subtasks, each paired with its corresponding conjunction. To exclude a completed subtask, we mask both the subtask tokens and their associated conjunction by replacing these tokens with a <mask> token when reconstructing the instruction.
>
> Weakness 3.3:
>
> Since our method works as a framework, the auxiliary loss is computed each time the function to compute the reinforcement learning loss from the baseline model is invoked. For example, in the BabyAI baseline, both the RL loss and our auxiliary loss are calculated over multiple rollouts, with the exact number of frames defined in the original baseline.
> We will also update the pseudo-code to help readers better understand the inner workings of our framework.
>
> Weakness 3.4:
>
> The exact function for the probability is as follows:
>
> $$prob = tanh(current frame/(max frames))$$
>
> where max_frame is a hyperparameter of RL models, determining how long the model is going to be trained for.
>
> Requested Changes 3:
>
> Thank you for your feedback. We have not explicitly tuned this hyperparameter for each task. However, due to time and resource constraints, we were unable to obtain the results during this time. Given sufficient time, we would be able to complete this evaluation.
>
> Requested Changes 4:
>
> The efficiency we refer to is sample efficiency, not time efficiency. Here, it is the same as the number of frames seen by the model. Many works on RL [3,4] consider the number of interactions between the agent and the environment—referred to as sample efficiency—as the primary challenge, rather than the overall training time. This is because interactions with the environment are inherently constrained by the environment’s properties and cannot be accelerated by increasing computational resources.
>
> [1] Chevalier-Boisvert, Maxime, et al. "BabyAI: A Platform to Study the Sample Efficiency of Grounded Language Learning." International Conference on Learning Representations.
>
> [2] Paischer, Fabian, et al. "Semantic HELM: an interpretable memory for reinforcement learning." CoRR, abs/2306.09312(2023).
>
> [3] Yang Yu. “Towards sample efficient reinforcement learning.” International Joint Conference on Artificial Intelligence. IJCAI, 2018.
>
> [4] Ladosz, Pawel, et al. "Exploration in deep reinforcement learning: A survey." Information Fusion 85 (2022): 1-22.

---

### Review · Reviewer_xPer · 2025-06-11

**Summary Of Contributions:**

This paper introduced a novel CAREL framework for instruction-guided reinforcement learning. This framework contained two crucial components: cross-modal auxiliary loss functions and instruction tracking mechanism. Experimental results demonstrated the effectiveness of the proposed framework over baselines in terms of the overall performance and sample efficiency.

**Audience:**

Yes

**Broader Impact Concerns:**

No concerns on the ethical implications of the work needs to be considered.

**Claims And Evidence:**

Yes

**Requested Changes:**

Please see above

**Strengths And Weaknesses:**

Strengths:

(1)  A novel CAREL framework based on cross-modal auxiliary loss functions and instruction tracking mechanism was introduced for instruction-guided reinforcement learning.

(2) Experiments on MiniGrid and BabyAI environments showed the effectiveness of the proposed modules in terms of the overall performance and sample efficiency.

Weaknesses:

(1) Some technical details can be further explained.
- It is stated that "as the auxiliary loss improves the similarity scores through time, and the improved similarities enhance the instruction tracking." This can be verified in the experiments to demonstrate the benefits of auxiliary loss and instruction tracking.
- It is stated that "one can constrain the probability of masking and relax this constraint gradually as the learning progresses". More discussion can be provided, e.g., how such constraints can be designed for the training process, and how it will prevent false positives, etc.

(2) More experiments and discussion can be conducted to verify the effectiveness of CAREL.
- The discussion of time consumption in subsection 3.1.2 is unclear. Why is it stated that the results in Table 2 don't show a substantial challenge? Furthermore, what is the "number of interactions with the environment needed for training" for CAREL in the experiments?
- Subsection 3.1.3 shows that the values above 0.01 in Figure 3 result in a decline in performance and 0.001 corresponds to the best performance. However, it is unclear whether smaller values of auxiliary loss coefficient can lead to improved performance.

---

> ### Author Response · Authors · 2025-06-30
>
> We sincerely thank you for your detailed and constructive review.
>
> Weakness 1.1:
>
> We have conducted a new experiment to show this effect in Appendix Section A.2.4. As shown in Figure 8, applying Instruction Tracking to our model (baseline + CAREL) results in a slightly higher improvement compared to its application on the original baseline.
>
> Weakness 1.2:
>
> We have provided additional explanation and examples for your information in the new implementation details subsection under the instruction tracking Section 2.2. We still reiterate them here for your convenience:
>
> We apply masking with a probability that increases based on the number of frames observed so far. Early in training, when the model has not yet learned strong representations and grounding, subtasks are rarely masked to avoid incorrectly masking those that are still incomplete (since masking of unaccomplished subtasks that are false positives may mislead the training process). As training progresses and representations improve, we increase the likelihood of applying masking accordingly.
>
> The exact function for the probability is as follows:
>
> $$prob = tanh(current frame/(max frames))$$
>
> where max_frame is a hyperparameter of RL models, determining how long the model is going to be trained for.
>
> Weakness 2.1:
>
> By the number of interactions, we mean each time the model sends an action to the environment and receives an observation. Here, it is the same as the number of frames seen by the model. Many works on RL [1,2] consider the number of interactions between the agent and the environment—referred to as sample efficiency—as the primary challenge, rather than the overall training time. This is because interactions with the environment are inherently constrained by the environment’s properties and cannot be accelerated by increasing computational resources.
>
> Weakness 2.2:
>
> Thank you for your feedback.
> The best-performing value mentioned in Subsection 3.1.3 should be 0.01 (the value 0.001 was a typographical error). Additionally, we have updated Figure 5 and included further results to show the sudden decline in performance for values below 0.001.
>
> References:
>
> [1] Yang Yu. “Towards sample efficient reinforcement learning.” International Joint Conference on Artificial Intelligence. IJCAI, 2018.
>
> [2] Ladosz, Pawel, et al. "Exploration in deep reinforcement learning: A survey." Information Fusion 85 (2022): 1-22.

---

### Review · Reviewer_Fzp8 · 2025-06-15

**Summary Of Contributions:**

This work explores enhancing existing language guided RL tasks with an multi-grained contrastive learning loss adopted from XCLIP deemed CAREL (Cross-modal Auxiliary Reinforcement Learning) framework. Authors claimed the loss should bring better cross modal grounding. In addition, an instruction tracking procedure is developed to track status of subtask and selectively mask out the completed tasks' instruction. Authors claimed both the loss and instruction tracking can benefit sample efficiency.

**Audience:**

Yes

**Claims And Evidence:**

No

**Requested Changes:**

1. Please provide discussion or additional analysis on "instruction tracking help the agent focus on the remaining tasks by preventing the repetition of completed sub-tasks." and on "improve cross-modal grounding between language instructions and environmental observations" or consider refine those claims.
2. For instruction tracking experiment, how is subtask being determined? an example would be helpful.
3. The time consumption analysis and the auxiliary loss coefficient can be fit into appendix for more room for additional discussion.
4. Experimental setting in consistent: instruction tracking experiment only using S_{E-W} to compute similarity, Vanilla CAREL is not using action embedding, I am not sure why the authors make such choice, as this would make it hard to know if the entire framework works. In addition, I notice the GoToSeq and OpenDoorOrderN4 experiments in Figure 2 and Figure 4 are done with different settings (the former not using action embedding, the latter uses action embedding), is there a reason for this?

**Strengths And Weaknesses:**

Strengths:
- One of the main methods is mainly adopted from a well recognized research work.
- Method description is well written and clear.
- Method is simple and model agnostic (it does ask for a language encoder and visual encoder to be already presented in a model)
- The method brings substantial sample efficiency as per the presented experiments.

Weakness:
- More experiments on settings other than BabyAI should be considered to prove the effectiveness of the method. Since it is mostly adopted, I believe diverse set of experiment should help enhance the evidence of the method working in language guide RL setting, hence the adoption itself can be valuable.
- There should be more analysis on if the added loss brings about better cross modal grounding rather than only claiming it to be true.
- "instruction tracking help the agent focus on the remaining tasks by preventing the repetition of completed sub-tasks." is also not evidently proved or discussed in experiments.
- Some experimental choices are confusing (see requested changes).

---

> ### Author Response · Authors · 2025-06-30
> **Answer**
>
> We sincerely thank you for your detailed and thoughtful review and deeply appreciate your acknowledgment of the clarity and comprehensiveness of our writing.
>
> Weakness 1:
>
> Thank you for your feedback; however, in addition to the BabyAI environment, we have also conducted experiments on the MiniGrid environment. For example, Semantic HELM (SHELM), which uses the powerful encoder of the CLIP model to learn strong embeddings, is evaluated in this environment. The results, presented in Figure 4, show a notable improvement in sample efficiency on this baseline.
>
> Weakness 2:
>
> The improvement is the direct result of contrastive loss [1] used in our methodology in Equation 12.
>
> The contrastive loss works by minimizing the distance between representations of related items (in this case, sequences of observations and the instruction) and maximizing the distance between representations of dissimilar ones. According to [1], this helps the model distinguish between different objects and their attributes, improving the sample efficiency by quickly identifying objects in the observation that are related to the instruction. Since we did not have access to frame annotations, we were unable to directly evaluate the grounding. However, given additional time, we could annotate the frames ourselves to enable a thorough assessment of the grounding.
>
> Weakness 3:
>
> We have provided additional explanation and examples for your information in the new implementation details subsection under the instruction tracking Section 2.2. However, we still reiterate a part of it here for your convenience:
>
> When a subtask is flagged as completed, instruction tracking replaces tokens associated with a subtask with <mask> tokens in the instruction. This means the model will no longer see that subtask in the instruction given to it and will have to focus on other subtasks.
>
> Requested Changes 1:
>
> Please refer to the new implementation details subsection under the instruction tracking Section 2.2.
>
> Requested Changes 2:
>
> We have provided the answer to this question in the new section. We still reiterate a part of it here for your convenience:
>
> In our case, the environments provide a clear and consistent instruction format, which enables us to split sentences into subtasks by using string matching to identify conjunctions for subtask separation.
>
> For instance, the “GoToSeq” environment uses the following structure:
>
> “go to a/the {color} {type}” + “and go to a/the {color} {type}” + “, then go to a/the {color} {type}” + “and go to a/the {color} {type}”
>
> Given this format, we tokenize the sentence and match the words “and” and “then” to identify and separate the subtasks.
>
> To extend this for environments without a clear format, Large Language Models such as GPT-4o can be used to break the instructions down to subtasks.
>
>
> Requested Changes 3:
>
> Thank you for your feedback. We have updated the paper.
>
> Requested Changes 4:
>
> Our auxiliary loss only focuses on representation learning and, as such, doesn’t need action embeddings. Conversely, Instruction Tracking (IT) makes use of the actions taken up to this point to predict the completion of a subtask; therefore, adding the action embeddings to the observations helps the model to better detect the completion of subtasks that are utilized in the IT mechanism. The differences between the GoToSeq and OpenDoorOrderN4 experiments in Figures 2 and 3 are also due to the CAREL+IT setting used in Figure 3, compared to the vanilla CAREL setting in Figure 2.
>
> To further investigate this, we conducted a CAREL+IT experiment on the GoToSeqS5R2 environment, comparing performance with and without action embeddings. As shown in the results (see Appendix  A.2.2), using action embeddings leads to a slight improvement, supporting their inclusion in the framework. Nonetheless, the exclusion of actions does not result in a substantial decrease in the performance of CAREL+IT, and so we can use a consistent setting in both CAREL and CAREL+IT by excluding actions. The results are depicted in Figure 6.
>
> When calculating the threshold in Equation 14, we can only use the $S_{E-W}$ or $S_{O-W}$ because we need the similarity of each word separately to be able to calculate a final score for each subtask. Between the two, we only used $S_{E-W}$ in our experiments for computational efficiency. The experiment in Figure 7 indicates that the $S_{E-W}$ alone suffices to reach a proper score for each subtask. (Refer to Appendix A.2.3)
>
> References:
>
> [1] Radford, Alec, et al. "Learning transferable visual models from natural language supervision." International conference on machine learning. PMLR, 2021.

---

### Decision · Action_Editor_ySSx · 2025-07-15

**Recommendation:** Accept as is

**Audience:**

Yes

**Audience Explanation:**

Instruction-guided reinforcement learning, instruction tracking, and cross-modality are all topics that a large part of the community is excited about.

**Claims And Evidence:**

Yes

**Claims Explanation:**

This paper introduces CAREL, a framework for instruction-guided reinforcement learning, which utilizes a contrastive learning loss and an instruction tracking mechanism to enhance cross-modal grounding, improve observation and instruction representations, and boost sample efficiency. Experimental results demonstrate that CAREL, combined with instruction tracking, significantly outperforms baselines in both overall performance and sample efficiency on two benchmarks.

Some concerns have been raised regarding the fact that the impact of some components of the solution was not empirically validated, but overall, reviewers thought the contribution had practical relevance, and the focus on sample efficiency was a positive aspect identified by the reviewers.